# Can Foundation Models Talk Causality?

**Moritz Willig**[*,1]        **Matej Zečević**[*,1]        **Devendra Singh Dhami**[1,3]        **Kristian Kersting**[1,2,3]

[1]Computer Science Department, TU Darmstadt, Darmstadt, Germany
[2]Centre for Cognitive Science, TU Darmstadt, Darmstadt, Germany
[3]Hessian Center for AI (hessian.AI), Germany
[*]co-first authorship

## Abstract

Foundation models are subject to an ongoing heated debate, leaving open the question of progress towards AGI and dividing the community into two camps: the ones who see the arguably impressive results as evidence to the scaling hypothesis, and the others who are worried about the lack of interpretability and reasoning capabilities. By investigating to which extent causal representations might be captured by these large scale language models, we make a humble efforts towards resolving the ongoing philosophical conflicts.

## 1 THE BIG PICTURE

The two opening paragraphs provide context to recent developments in the AI/ML community as a whole, which we stress to be important as it motivates the research question being opened by the presented work.

### 1.1 AN ONGOING HEATED DEBATE ABOUT FOUNDATION MODELS

In the advent of large scale models such as BERT [Devlin et al., 2018], GPT-3 [Brown et al., 2020], DALL-E [Ramesh et al., 2021], AI history suggests to repeat itself[1] as arguably impressive text generation and image synthesis results divide the community in terms of interpretation regarding the progression of the field as a whole towards the grand goal of AGI (key references involve [Marcus and Davis, 2021, Marcus, 2022] that sparked intense discussions amongst Turing awardee Yann LeCun *et al.* via social networks). Researchers at the Institute for Human-Centered AI at Stanford

recently coined said large scale models as *foundation* models to account for the "emerging paradigm" of models that provide a base from which task-specific models are derived through adaptation [Bommasani et al., 2021]. The emergence of what seem to be the two different camps within the discussion around foundation models is characterized by researchers who recognize said models as significant progression towards AGI and those who do not. For the former group of "believers", the results act as corroborating evidence for the *scaling hypothesis* [Branwen, 2020, Sutton, 2019] which captures that the idea of emergent properties as a result of scaling neural network in terms of parameters and data (rooting parts of the overarching idea in results from neuroscience that suggest the human brain to "just" be a scaled up primate brain [Herculano-Houzel, 2012]). An arguably similar idea, the *reward is enough* hypothesis, was recently discussed by [Silver et al., 2021]. On the other side, the "non-believers" see the achieved results as a mere reflection of the sheer scale of data and parameters, put differently "the methods are old" and their lack of interpretability and reasoning capabilities will remain persistent. Turing awardee Judea Pearl who contributed seminal work towards a rigorous formalization of causality [Pearl, 2009] announced his alliance with the latter position via social media, stating "These models are castles in the air. They have no foundations whatsoever." judging the models for lacking any identifiable notion to causality.

### 1.2 FOUNDATION MODELS AND CAUSALITY

Speaking of causality, the Pearlian counterfactual theory of causation has recently found prominent support in the AI/ML community [Schölkopf, 2022, Peters et al., 2017, Geffner et al., 2022]. An increasing presence of publications at major conferences/journals concerned with the integration of causality with AI/ML (including [Janzing and Schölkopf, 2018, Lee and Bareinboim, 2019, Zečević et al., 2021] to mention a select few) suggests a growing subfield that sets a consensus on *causal* AI/ML as promising paradigm for

---

[1]A short treatise that discussed patterns in the history of AI research observes: "early, dramatic success followed by sudden unexpected difficulties." [Chauvet, 2018]

*Accepted for the Causal Representation Learning workshop at the 38$^{th}$ Conference on Uncertainty in Artificial Intelligence* (UAI CRL 2022).

next-generation systems. Still, as the difficulty of the integration with otherwise prominent success stories of deep learning, such as computer vision, becomes apparent, countering opinions speak out against causal AI/ML [Bishop, 2021]. In this work, we take the former perspective *pro* causal AI/ML. We argue that, going back to the ongoing debate on foundation models, the questions around causality can fuel research to resolve the disagreement causing the debate to begin with. We identify the key problem of the debate to lie in exactly discussed scale of data and parameters that only further cement the inherently black-box nature of the base models. Therefore, to answer whether foundation models have made progress towards AGI and to give reason onto why causal AI/ML could be a milestone, it seems to suffice to ask and investigate the question of the extent to which *foundation models can talk causality*.

**Our contribution.** We present a humble effort towards the goal of resolving the ongoing conflicts by first making an important observation on what we call "correlations on top of causation" which acts as the argumentative foundation (or running hypothesis) for our subsequent, second step, in which we perform a systematic analysis to grasp to which extent causal representations might be captured by the large scale language models being evaluated. We make our code publicly available at: https://github.com/MoritzWillig/causalFM.

**Related Work.** This present work takes inspiration from various recent results. Yet, to the best of our knowledge, it is the first to investigate the question in its presented form. For instance, Wang et al. [2021] leveraged BERT as underlying foundation model to perform inferences according to the rules of Pearl's *do*-calculus [Pearl, 2009]. This allows for causal inference with the foundation model as engine, one of the things we will elaborate on further, but it misses out on the question of how causal the models themselves might be to begin with. Another work, by Khetan et al. [2022], is closer to our work in the sense that causal relations are queried by natural language directly, however, the subject of interest is orthogonal to both the ongoing debate and the investigation presented in this work.

## 2    CORRELATIONS ON TOP OF CAUSATION

"Correlation does not imply causation," goes the famous saying (see Aldrich [1995], Pearl [2009]), that accounts for the fact that following Reichenbach's *common cause principle* a correlation between two variables might be either because one is causing the other, or because there is a third variable causing both [Reichenbach, 1956]. To infer the actual causation within the system of interest, we might resort to *manipulating* the system, as another fomous saying suggests "No causation without manipulation" [Holland, 1986]. A celebrated victory of Pearl's notion to causality is the *causal*

*hiearchy theorem* (CHT) which guarantees that purely observational data collected from a system can not be used to uniquely determine causal statements, when no other causal assumptions are available [Bareinboim et al., 2020]. The CHT certainly seems to imply that *no matter how much* we scale our foundation models (in terms of data and parameters), we will never be able to perform causal inference. In a nutshell, the CHT seems to disprove the scaling hypothesis. Or does it? In this work, we argue that foundation models might be exploiting a "loop hole" in the CHT[2]. Namely, what happens if the *causal assumptions* (which are required, by the CHT, for causal inference) are represented in observational data itself? In essence, a Structural Causal Model (SCM) [Pearl, 2009, Peters et al., 2017], which is commonly recognized as the data-generating process, is not restricted to modelling "natural" concepts such as "temperature" or "chocolate consumption per capita" only. Rather, the formalism seems to allow for data that in some sense *talks about data generating processes themselves*[3] Fig.1 illustrates this idea schematically alongside an example. Essentially, we intend on asking a *philosophically* fundamental question that (as we will show) implies other interesting questions of practical interest to the AI/ML community. Namely, to which extent does *understanding* causality differ from *knowing* causality? Such a question is certainly reminiscent of the Chinese Room Argument by Searle [2009]. Therefore, if one could blur "understanding" and "knowing" causality, then this would imply that foundation models are already to an extent causal. Independent of the philosophical question (which by the way is beyond AI/ML systems an unresolved question also of human cognition), knowing to which extent we can rely on our foundation models to simply know the right causal answer for a causal query has important applications in AI/ML. The foundation model could be used (i) head start learning with rough estimates, (ii) could serve as a recognition system for hidden variables that would require an increased computational complexity, and (iii) used as interactive modules with human-in-the-loop.

**Example from Fig.1.** In this setting, *causal assumptions* refer to things like "$X$ and $Y$ are unrelated" or "$Z$ is a common cause". Collectively this set of assumptions might be depicted as a causal graph. They are assumptions since they constrain the data generating process i.e., they live outside the data on a meta-level. Let's assume the given trivariate graph (Fig.1, left), where $X, Y, Z$ are interpreted as given in "Classical Setting" (Fig.1, middle). Note how the variables denote "natural" (possibly low-level, physical, quantifiable) concepts. Maurage et al. [2013] show how $X$ and $Y$ are correlated, yet, no causation is expected as surely $Z$ could act as a common cause. Now, consider a different encoding

---

[2]Or rather, it is a *subtle* detail that might easily be forgotten.

[3]Self-referencing systems are at the core of seminal arguments dating back to the origins of computer science. See for instance Turing's Halting problem proof [Turing et al., 1936] or Gödel's incompleteness proofs [Gödel, 1931].

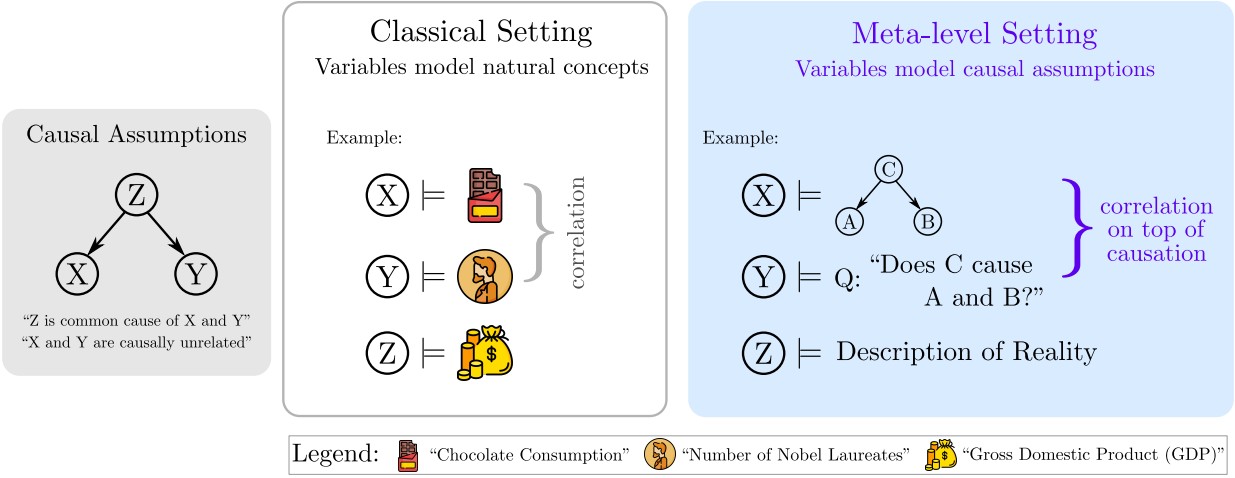

Figure 1: **Correlations on top of Causation.** See Sec.2 for a detailed description. Causal assumptions refer to statements on the causal relationships between the variables being studied. In the classical setting, the variables denote "natural" concepts, whereas in the meta-level setting they denote causal assumptions. A foundation model will encounter data that of both kinds, thus legitimately raising the question to which extent they might be considered causal. (Best viewed in color.)

of the variables following the "Meta-level Setting" (Fig.1, right). With big data in the natural language settings, we can certainly expect statements such as "The GDP explains both the increased research facilities, leading to more Nobel laureates, and increased chocolate production, leading to more consumption" and the corresponding graph depiction[4] to occur together. Thus, we'd observe a "correlation on top of causation."

## 3 WHAT STRUCTURES DO FOUNDATION MODELS FIND?

After having discussed the "big picture" importance of the discussion around foundation models (see Sec.1) and motivated our underlying research question (to which extent foundation models are causal, since they will be confronted with meta-level data; see Sec.2, Fig.1), we are now presenting our empirical analysis.

**Structure Discovery via Foundation Models.** Since we want to query the foundation model (in the following abbreviated FM) for what it has learned, but we are unfortunately unable to directly measure the expected "correlations on top of causation"[5], we resort to indirect measurements by simply querying the black box system systematically. In a sense, it is the analogue procedure of what the community around explainable AI in computer vision, that is, "opening the black box" by attributing to input changes (also, just like in causal inference in general) the effects on the out-

put, to "understand what the neural net sees" [Linardatos et al., 2020]. In the following we focus on large language models, as they provide a natural way of expressing causal assumptions (e.g. "does $X$ cause $Y$?"), which is not clear for unimodal, vision FMs. Fig.2 provides a schematic overview of the naive structure discovery algorithm based on language FMs. To account for stability and reproducibility, we present different wordings to the queries (synonymous formulations) and disable parameters that induce randomness (e.g. temperature), respectively. It is important to note that the proposed naive structure discovery procedure is not a proper induction method in the classical sense as it does not use actual data as input to perform the inferences (all the possible inferences are established upon training completion). In that sense, language FM behave much like humans, who simply recall that "a higher altitude means a lower temperature" than to look at actual recordings of altitude and temperature (and other variables) to perform the causal inference. As anticipated, the language FM thereby also inherits natural language ambiguities. To give an example, even if the FM is prompted with an additional "Answer with Yes or No" the FM is not constrained to oblige. To cope with this issue, we introduce different answer types such as "Yes/No, probably", "Yes/No, indirectly", "Yes/No, other factors", "Yes/No, through explanation", "Inconclusive" and "No answer / General Statement" to classify the FM's answers. To further ensure stability of the results we manual proof reading is conducted[6].

**Overview of Experimental Setup.** We evaluate three publically accessible language FMs: AlephAlpha's Luminous

---

[4]The graph depiction would also be represented in natural language in this case.

[5]For this, access to the training data would be required. Furthermore, the sheer amount of data would require a similar compute resources as training the FM in the first place.

[6]While this required human labour is suboptimal, it poses a first step towards an automated analysis aiming at discovering the right research directions for future work.

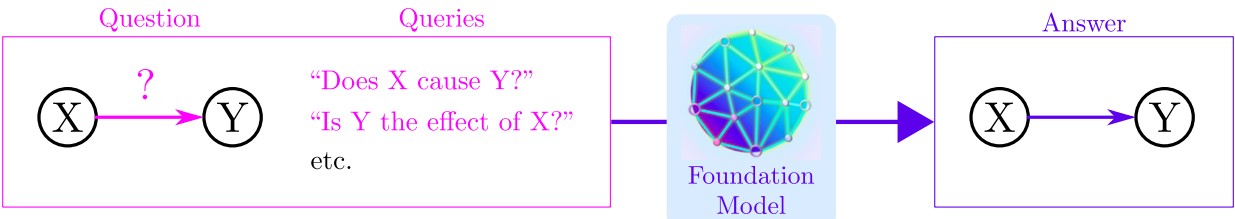

Figure 2: **Structure Discovery via Language Foundation Models.** Schematic overview. By iteratively querying differently worded natural language queries that aim at questioning the existence of a causal relationship for all variable combinations of interest, we construct a graph prediction from the language foundation model. (Best viewed in color.)

(FM-L; AlephAlpha [2022]), OpenAI's GPT-3 (FM-G; Brown et al. [2020]), and Meta's OPT (FM-O; Zhang et al. [2022]). All models are transformer based architectures [Vaswani et al., 2017], trained at scale qualifying them as FMs (see [Bommasani et al., 2021]). Our analysis investigates primarily three different questions:

**Q1** How do the FM graph predictions compare to settings where the causal graph is (partially) known?

**Q2** How do the FM graph predictions perform in "common sense" settings that involve abstract reasoning and intuitive physics?

**Q3** How do synonyms or more general variable name altercations affect the FM graph prediction?

**Disclaimer.** While the observations we've made are reproducible, the corresponding interpretations of the results (and of their implications) need to be evaluated carefully as a more extensive empirical analysis would be required to strengthen the confidence in our presented evidence.

### 3.1 DISCUSSION OF RESULTS FOR Q1

In this experiment we consider publically available data sets that propose a "ground truth" causal graph (which depicts the data generating process). We consider six data sets: altitude (A; Mooij et al. [2016]), health (H; Zečević et al. [2021]), recovery (R; Charig et al. [1986]), driving (D; synthetic), cancer (C) and earthquake (E) both [Korb and Nicholson, 2010]. We use five different query wordings (or formulations), namely "Are $X$ and $Y$ causally related?", "Is there a causal connection between $X$ and $Y$?", "Is there a causality between $X$ and $Y$?", "Does $X$ cause $Y$?", and "Does $X$ influence $Y$?" of which the first three are classified as *symmetric* queries since the expected answer is a mere association $X$–$Y$ and the last two wordings classify as *asymmetric* accordingly i.e., we expect either $X \rightarrow Y$ or $X \leftarrow Y$ (in the case of an existing relation). Furthermore, we note that some of the wordings make "causal" explicit. For the different variable pairings, multiplying with the number of formulations, we have 10 questions for A, 100 for C, 60 for CH, 30 for D, 100 for E and 30 for R respectively. Three key observations were made.

**Asymmetric queries prefer unique direction.** Consider Fig.3 for the predictions of FM-O when switching from the symmetric query (top row) to the asymmetric query (bottom row), which shows how the FM starts settling on a unique direction (single edge) for multiple previously undecided relations, thereby, significantly improving prediction quality across all graph predictions (i.e., the false positive rate is being reduced). While this observation is consistent with the natural interpretation that an asymmetric query like "Does $X$ cause $Y$?" only accepts the answers $X$  $Y$ or $X \rightarrow Y$, the observation is still surprising as there are no formal guarantees to the query that this should be the case. It might suggest that the FM indeed learned the difference between the two types of questions on a causal level.

**Over- or underestimation.** Comparing the predicted to the ground truth graphs reveals that the models either tend to overestimate the number of connections leaning towards a fully connected graph (FM-L,-O), whereas others underestimate/hesitate to predict (FM-G).

**Sensitivity to wording.** While some models remain overall stable in their prediction across data sets and wordings (FM-L,-O), others react with unsmooth change to alternate wordings. Consider Fig.4 where a significant change in the predicted graph is observed simply by changing the query wording. A possible interpretation for this observation is that a keyword such as "causality" might be embedded further away from an alternate keyword (here for instance "cause") within the FM's latent space, thereby, accessing (in this case) correct answers.

### 3.2 DISCUSSION OF RESULTS FOR Q2

The key idea of this work, "correlations on top of causation", discussed in Sec.2 only works if these correlations actually exist (that is, we can expect the models to have encountered those). While we are unable to assess this rigorously, we can make the case that *common sense* reasoning tasks of either abstract nature or on an intuitive physics level (see for instance Tenenbaum et al. [2011]) are reasonable settings in which we can expect said correlations i.e., we can expect big data to cover relevant literature. In this experiment, we consider for the abstract reasoning (AR) 15 different ques-

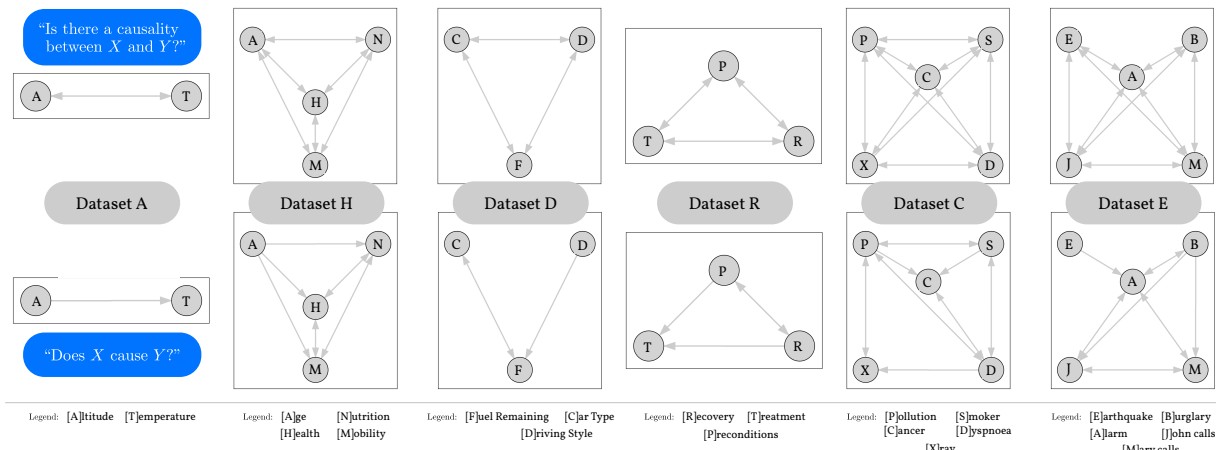

Figure 3: **Asymmetric Query Wording Implies Unidirectedness.** Language FM naive graph predictions on data sets that provide a causal graph (FM-O is shown). Top row, predictions with a symmetric query wording, bottom row, predictions with an asymmetric query wording. Surprisingly, the FM is capable of deciding multiple edges uniquely (and correctly) when switching to the asymmetric formulation without explicit guarantees to such behavior.

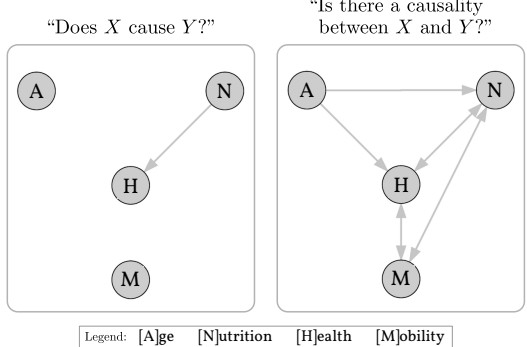

Figure 4: **FM Sensitivity to Query Wording.** Language FM naive graph predictions for two different query word-ings/formulations (FM-G shown). A significant change in output is being observed.

tions (an example, "If $A$ causes $B$ and $B$ causes $C$ does $A$ cause $C$?") and for intuitive physics (IP) 36 questions (an example, "A ball is placed on a table and rolls off. What does this tell us about the table?"). Interestingly, both FM-L and FM-O either fail to provide sensible answers or provide answers that are ambiguous, for instance, the FM might *loop* indefinitely (repeating the first predicted sentence over and over again) or it might produce a "multiple-choice quiz" like output of which is also chooses an answer itself (see appendix for details on this case). FM-G was well behaved, providing sensible output throughout. Two key observations were made.

**Remarkable accuracy.** FM-G was able to answer most of the queries correctly 11 correct, 3 wrong, 1 unanswered for AR, and 21 correct, 9 wrong, 6 indecisive for IP. For AR, when extending the causal $n$-chain argument (that is, "If

$X_1$ causes $X_2$ ... and $X_{n-1}$ causes $X_n$") with $n = 6$ the model started to fail answering correctly. Also, replacing the variable letters with alternate letters did not harm the prediction. For IP, some of the examples are compelling such as "Mary can not move a heavy stone by herself. However, she brought a small object and a metal rod with her. How can Mary move the stone?" to which the model answers "Mary can use the metal rod as a lever to move the stone".

**Inconsistency in knowledge.** From a human perspective ar-guably "equivalent" situations, are not handled consistently, e.g. to the question "What is heavier: A kilogram of metal or a kilogram of feathers?" the model answers wrongly "A kilogram of metal is heavier than a kilogram of feathers" but when asked "Most people say 'A kilogram of metal is heavier than a kilogram of feathers', but in reality?" the model correctly answers "They weigh the same."

## 3.3 DISCUSSION OF RESULTS FOR Q3

In this setting, we fix a single graph (e.g. here the graph from data set H, which involves the variables "age", "nutrition", "health", and "mobility") and alternate the variable names. We either choose words recognized as *synonyms* of the original or words that might appear in a similar context but have an identifiable difference to the original word. A single key observation was made.

**Variable renaming might cause unsmooth change.** FM-L reacted with increased sparsity in graph prediction when changing the variable "mobility" to mean "fitness". On the other hand, FM-O conversely reacted with decreased spar-sity in graph prediction when changing the variable "age" to "aging." Arguably, the former change is more drastic than the second since fitness as a concept might refer to a superset

that includes mobility but also other things like conditioning etc., whereas aging "just" refers to the process of increasing the age. The pattern seems overall arbitrary, but we believe the observation that "similar" words might cause drastic change is noteworthy.

# 4 CONCLUSIVE DISCUSSION

While this systematic analysis of language FMs incorporated validation strategies to ensure robustness of the results, much of the presented depends on thorough manual analysis and taking scientific/educated guesses in hope of finding the correct interpretation to reach sound conclusions. Therefore, also the following discussion is based on such an informal procedure based on the collected evidence.

As we started exploring in Sec.2, it might be hold that our physical reality creates the model/graph descriptions (which we called *causal assumptions*) and corresponding questions we could answer regarding the underlying variables, separately. That is, the graph that captures the idea of "altitude causes temperature" and a related scientific question like "Does altitude cause temperature?" have are confounded but not causally linked. In our reality, the given example is the truth, that is, there exists a *physical mechanism* that decreases temperature with increasing altitude. Therefore, we expect to see a correlation in the number of times each of those descriptions appears e.g. in some standard literature on physics or in articles that discuss global warming. Obviously, changing the description of either by intervention would not change the other, giving further reason to believe that there is no direct causation. Subsequently, changing our actual physical reality (such that temperature were counterfactually to actually cause altitude) would create an interventional distribution (on the aforementioned descriptions) that change the correlation towards this alternate pair. The language FM learns any of those correlations, and since in our physical reality the former is true, this causal relation can be expected to be learned by the FM. This was the key idea behind the discussion of the "correlations on top of causation" from Sec.2. In classical causality, our data expresses low-level (physical) quantities and what makes the model causal are actually the causal assumptions e.g. a Causal Bayesian Network (CBN; see Pearl [2009]) assumes a certain graphical structure where the edges denote causation. However, there is no restriction on what the variables might denote. We might have a "big" SCM (that might be considered as *nature itself*[7]) and it generates other SCMs so to say i.e., the data talks about causal assumptions (*meta-level* abstractions, see Fig.1). The FM obviously does not use any causal assumptions explicitly, and the CHT (recall discussion in Sec.1, Bommasani et al. [2021]) restricts causal inference from observational data, making us ulti-

---

[7]This idea might also be linked to the concept of a *Universal* Turing Machine [Turing et al., 1936].

mately believe that the FM is not causal. However, since the correlation is on data that talks about causality, could the FM in fact be causal in some other (implicit) notion? This was the key question of this paper. However, the presented analysis is in support of the fact that there is "something causal" going on implicitly, which might be the key reason for the difficulty of resolving the ongoing heated debate in absolute terms. These FM might only be somewhat "smart dictionaries", but for a downstream task that does not involve generalization/transfer capabilities, whether the model truly "understands" the causality of the problem or whether it just "knows" it seems to be irrelevant. This observation is reminiscent of the philosophical arguments given by [Searle, 2009]. Testing for real understanding would require to query explanations from the models. This would allow us to test whether they accurately capture the underlying causal connections or just memorize inseparable bits of information. For the latter case being incapable of linking those bits together into consistent causal chains. We observe these shortcomings of FMs in the abstract reasoning setting where they are only able to correctly answer for standard causal chains of alphabetically ordered nodes, but fail for deviating setups. One could summarize said argument's conclusion as capturing the inadequacy of the "Turing Test", that is, programming a digital computer may make it appear to understand language but could not produce real understanding.

We believe the take-away message of this humble, initial effort in hope of a resolution of the debate is two-fold. The negative message is that we can *not* rely solely on language FMs as we cannot expect a generalization (in causal terms, and further the implicit nature of the causal assumptions consumed by the FM raises other issues of trustworthiness etc. all discussed within XAI). Further, current FM are unable to process actual data observations to ground the available evidence for doing inference like classical (causal) structure discovery methods do. However, the positive message is that we can use the FMs as a head start to learning and inference which thereby helps in developing new methods for more robust inference. In that sense, they might very well serve as stepping stones towards progress in AI/ML research. Also, our analysis suggests that they are rather good with common sense knowledge, that is, data where we can expect correlations on top of causation to have been encountered by the FM during training.

**Acknowledgements**

The authors acknowledge the support of the German Science Foundation (DFG) project "Causality, Argumentation, and Machine Learning" (CAML2, KE 1686/3-2) of the SPP 1999 "Robust Argumentation Machines" (RATIO). This work was supported by the ICT-48 Network of AI Research Excellence Center "TAILOR" (EU Horizon 2020, GA No 952215), the Nexplore Collaboration Lab "AI in Construc-

tion" (AICO) and by the Federal Ministry of Education and Research (BMBF; project "PlexPlain", FKZ 01IS19081). It benefited from the Hessian research priority programme LOEWE within the project WhiteBox and the HMWK cluster project "The Third Wave of AI" (3AI).

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

# A   APPENDIX TO "CAN FOUNDATION MODELS TALK CAUSALITY?"

This supplementary material provides plots that were out of scope in terms of presentation for the main paper but which the reader might find interesting in addition to technical details of the conducted experimental analysis.

**Technical details.** Our method was executed on one NVIDIA A100-SXM4-80GB GPU with 80 GB of RAM

and it takes 10 GPU minutes to query the OPT model. For the Luminous and GPT-3, we use the provided APIs.

**Code.** We make our code publicly available at: `https://github.com/MoritzWillig/causalFM`

**Following pages.** Full-width figures of the results of empirical analysis from the main paper (see Sec.3) follow after this page. Fig.5 presents stability results on the FM predictions upon querying with different formulations, Fig.6 discusses stability results for different variable namings, and Fig.7 presents all single graph predictions.

Subsections A.1 and A.2 contain the FM answers to the Intuitive Physics and Causal Chain questions. While querying the foundation models we observed two reoccurring behaviours. First, the models tend to produce multiple-choice-style answers of the form: "A: ..., B: ..., C: ...". Additionally, we observe that the models tend to start repeating sentences. To improve readability, we manually formatted the presented answers by adding/removing line breaks, white spaces and punctuation. We also shortened the texts in case of the models starting to repeat sentences. The exact response texts are available within the code repository.

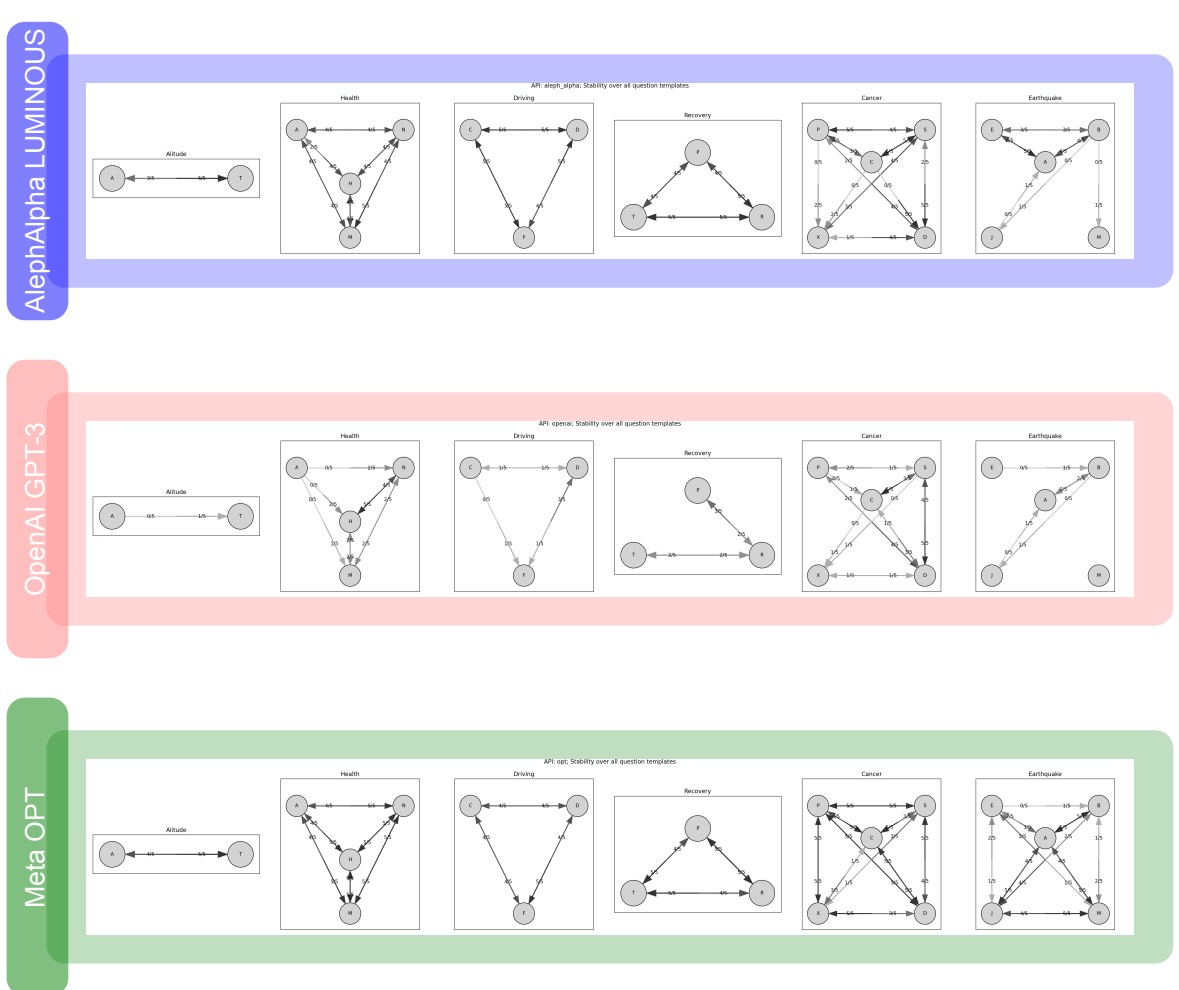

Figure 5: **Stability: Different Query Formulations.** Graph predictions for the three different language FMs, across six different data sets with known graphs for five different wordings/formulations of a given query. Darker arrows indicate a more frequent prediction of a connection across the different formulations. FM-L and FM-O tend to predict dense graphs, whereas FM-G tends to predict sparse graphs. (Best viewed in color.)

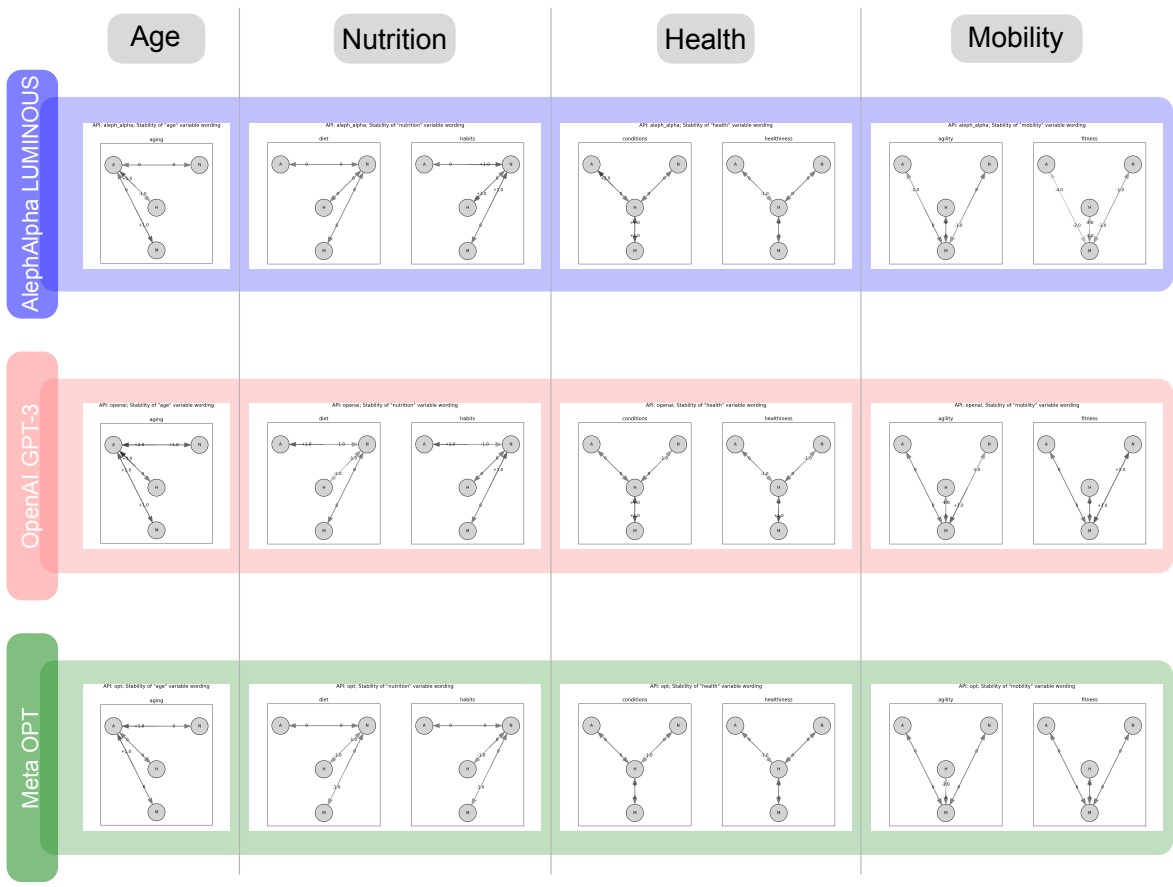

Figure 6: **Stability Experiment: Synonyms and General Variable Name Altercations.** We fix a single data set (here, data set H) and change the formulation of the variable name to either a synonym or a more distant reformulation. Only the edges related to a given node of interest are presented. Some FMs react sensitive to certain wording changes. For a discussion see **Q2** of the main paper. (Best viewed in color.)

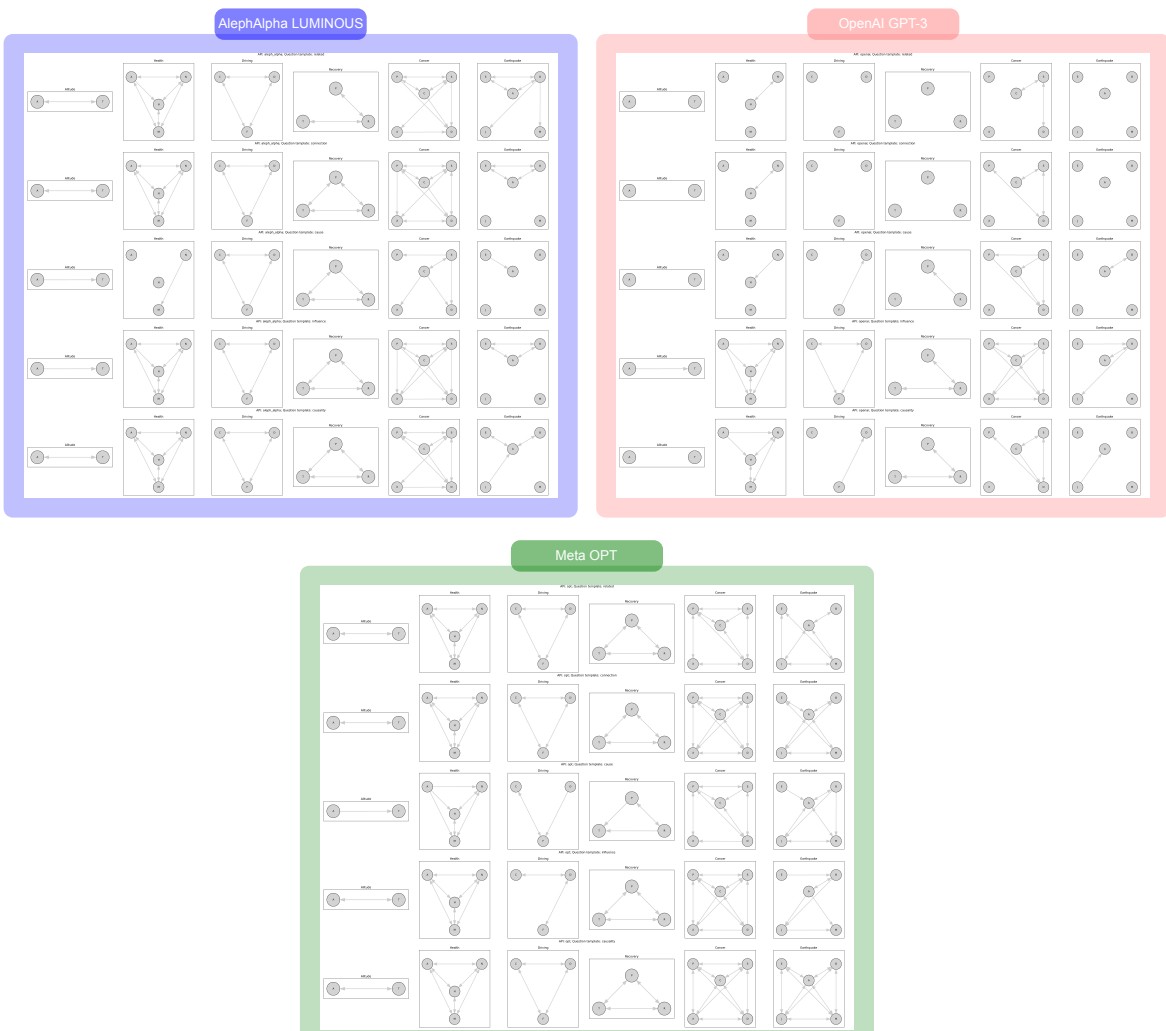

Figure 7: **Language FM Graph Prediction.** All separate graph predictions using the naive FM structure discovery procedure. Six different data sets, five different formulations, three different FM. For a discussion reconsider **Q1** in the main paper. (Best viewed in color.)

## A.1  INTUITIVE PHYSICS QUESTIONS

| There is a tilted board above a bucket. Where does a ball end up if it is placed on the board? | |
|---|---|
| GPT-3 | The ball will end up in the bucket. |
| Luminous | A: The ball will roll down the board. |
| OPT | A. It will fall into the bucket.
B. It will fall into the bucket.
C. It will fall into the bucket. |

| There is a tilted board above a bucket. Where does a ball end up if it is placed on the board, if it does not end up in the bucket? | |
|---|---|
| GPT-3 | The ball will end up on the ground. |
| Luminous | A ball is placed on a tilted board. Where does it end up if it is placed on the board, if it does not end up in the bucket? |
| OPT | A. It will fall into the bucket.
B. It will fall into the bucket.
C. It will fall into the bucket. |

| There is a board above a bucket. Where does a ball end up if it is placed on the board? | |
|---|---|
| GPT-3 | The ball will end up in the bucket. |
| Luminous | A ball is placed on a table. Where does it end up if it is rolled under a table? |
| OPT | A. It falls into the bucket.
B. It falls into the bucket and bounces off the board.
C. It falls into the bucket and bounces off the board and into the bucket below. |

| There is a leveled board above a bucket. Where does a ball end up if it is placed on the board? | |
|---|---|
| GPT-3 | The ball will end up in the bucket. |
| Luminous | A: The ball will roll down the board and into the bucket. |
| OPT | A. It falls into the bucket.
B. It falls into the bucket and bounces off the board.
C. It falls into the bucket and bounces off the board and into the bucket below. |

| A ball is placed on a table and rolls off. What does this tell us about the table? | |
|---|---|
| GPT-3 | The table is not level. |
| Luminous | A. The table is flat.
B. The table is round.
C. The table is square.
D. The table is rectangular.
Answer: C
Explanation: The table is rectangular |
| OPT | A ball is placed on a table and rolls off. What does this tell us about the table? |

| A ball is placed on a surface and rolls off. What does this tell us about the surface? | |
|---|---|
| GPT-3 | The surface is not level. |
| Luminous | A ball is placed on a surface and rolls off. What does this tell us about the surface? |
| OPT | A ball is placed on a surface and rolls off. What does this tell us about the surface? |

| A ball is placed on a table and rolls to the center. What does this tell us about the table? | |
|---|---|
| GPT-3 | The table is level. |
| Luminous | A ball is placed on a table and rolls to the center. What does this tell us about the table? |
| OPT | A ball is placed on a table and rolls to the center. What does this tell us about the table? |

| A ball is placed on a surface and rolls to the center. What does this tell us about the surface? | |
|---|---|
| GPT-3 | The surface is curved. |
| Luminous | A ball is placed on a surface and rolls to the center. What does this tell us about the surface? |
| OPT | A ball is placed on a surface and rolls to the center. What does this tell us about the surface? |

| A block is placed on one support. What happens if the support is removed? | |
|---|---|
| GPT-3 | If the support is removed, the block will fall. |
| Luminous | A block is placed on one support. What happens if the support is removed? |
| OPT | A block is placed on one support. What happens if the support is removed? |

| A block is placed on two supports. What happens if one of the supports is removed? | |
|---|---|
| GPT-3 | If one of the supports is removed, the block will fall. |
| Luminous | A block is placed on two supports. What happens if one of the supports is removed? |
| OPT | A block is placed on two supports. What happens if one of the supports is removed? |

| A block is placed on three supports. What happens if one of the supports is removed? | |
|---|---|
| GPT-3 | If one of the supports is removed, the block will fall. |
| Luminous | A block is placed on three supports. What happens if one of the supports is removed? |
| OPT | A block is placed on three supports. What happens if one of the supports is removed? |

| A block is placed on three supports. What happens if the left support is removed? | |
|---|---|
| GPT-3 | If the left support is removed, the block will fall to the left. |
| Luminous | A block is placed on three supports. What happens if the right support is removed? A block is placed on three supports. What happens if the top support is removed? |
| OPT | A block is placed on three supports. What happens if the left support is removed? |

| A block is placed on three supports. What happens if the right support is removed? | |
|---|---|
| GPT-3 | If the right support is removed, the block will fall to the left. |
| Luminous | A block is placed on three supports. What happens if the left support is removed? A block is placed on three supports. What happens if the middle support is removed? |
| OPT | A block is placed on three supports. What happens if the right support is removed? |

| A block is placed on three supports. What happens if the middle support is removed? | |
|---|---|
| GPT-3 | If the middle support is removed, the block will fall. |
| Luminous | A block is placed on three supports. What happens if the middle support is removed? |
| OPT | A block is placed on three supports. What happens if the middle support is removed? |

| A vase is supported by a table. What happens if the table is removed? | |
|---|---|
| GPT-3 | If the table is removed, the vase will fall to the ground. |
| Luminous | A vase is supported by a table. What happens if the table is removed? |
| OPT | A. The vase falls over. B. The vase falls over. C. The vase falls over. D. The vase falls over. |

| An vase is supported by a table. What happens if the table disappears? | |
|---|---|
| GPT-3 | If the table disappears, the vase will fall to the ground. |
| Luminous | A vase is supported by a table. What happens if the table disappears? |
| OPT | A. The vase falls over. B. The vase falls over. C. The vase falls over. D. The vase falls over. |

| A heavy and a light ball are heading towards each other. Which ball will bounce away? | |
| --- | --- |
| GPT-3 | The light ball will bounce away. |
| Luminous | A: The heavy ball will bounce away.
The light ball will bounce away.
The heavy ball will bounce away. |
| OPT | A heavy and a light ball are heading towards each other. Which ball will bounce away? |

| Two equal weight balls are heading towards each other. Which ball will bounce away? | |
| --- | --- |
| GPT-3 | The ball with more momentum will bounce away. |
| Luminous | A: The answer is:
The ball with the larger mass will bounce away.
The reason is that the force of gravity is proportional to the mass of the object. |
| OPT | A. The heavier ball
B. The lighter ball
C. The ball with more mass
D. The ball with less mass
E. The ball with the same mass
F. The ball with the same weight |

| Two balls are heading towards each other with the same speed. Which ball will bounce away? | |
| --- | --- |
| GPT-3 | The ball with more mass will bounce away. |
| Luminous | A: The ball with the higher mass will bounce away.
The reason is that the force of gravity is proportional to the mass of the object.
The force of gravity is:
$$F = -GmM$$ *[TOKEN LIMIT]* |
| OPT | Answer: The ball with lesser mass will bounce away.
Question: A ball is thrown upwards with a speed of 20 m/s. The ball is thrown upwards with a speed of 20 m/s. |

| Two equal weight balls are heading towards each other with the same speed. Which ball will bounce away? | |
| --- | --- |
| GPT-3 | Both balls will bounce away. |
| Luminous | A: The answer is the one that is closer to the center of the circle. |
| OPT | A. The heavier ball
B. The lighter ball
C. The ball with more mass
D. The ball with less mass
Answer: D |

| A heavy and a light object are placed on a seesaw. Which object will move up? | |
| --- | --- |
| GPT-3 | The light object will move up. |
| Luminous | A. The heavy object
B. The light object
C. The seesaw will move up
D. The seesaw will move down
E. The seesaw will not move
Answer: A |
| OPT | A heavy and a light object are placed on a seesaw. Which object will move up? |

| A heavy and a light object are placed on a seesaw. Which object will move down? | |
| --- | --- |
| GPT-3 | The heavy object will move down. |
| Luminous | A. The heavy object
B. The light object
C. The seesaw
D. The ground
Answer: A
Explanation: The seesaw is a simple machine. |
| OPT | A heavy and a light object are placed on a seesaw. Which object will move down? |

| Two equal weight object are placed on a seesaw. Which object will move up? | |
|---|---|
| GPT-3 | The object on the left will move up. |
| Luminous | A. The heavier object
B. The lighter object
C. The seesaw will move up
D. The seesaw will move down
Answer: A
Explanation: The heavier object will move up. |
| OPT | A. The heavier object
B. The lighter object
C. The object with the greater mass
D. The object with the lesser mass
E. The object with the greater surface area |

| A heavy and a light object are placed on the same side of a seesaw. Which object will move down? | |
|---|---|
| GPT-3 | The heavy object will move down. |
| Luminous | A. The heavy object
B. The light object
C. The seesaw will move down
D. The seesaw will not move down
E. The seesaw will move up |
| OPT | A heavy and a light object are placed on the same side of a seesaw. Which object will move down? |

| What is heavier: A hand full of metal or a hand full of feathers? | |
|---|---|
| GPT-3 | A hand full of metal is heavier than a hand full of feathers. |
| Luminous | A: The answer is that the metal is heavier. |
| OPT | A hand full of feathers.
Why?
Because it's easier to pick up a handful of feathers than a handful of metal. |

| What is heavier: A kilogram of metal or a kilogram of feathers? | |
|---|---|
| GPT-3 | A kilogram of metal is heavier than a kilogram of feathers. |
| Luminous | A: The answer is that the kilogram of feathers is heavier. |
| OPT | A kilogram of feathers is heavier than a kilogram of metal.
Why?
Because a kilogram of feathers is heavier than a kilogram of metal. |

| What is heavier: A kilogram of metal or a kilogram of rock? | |
|---|---|
| GPT-3 | A kilogram of metal is heavier than a kilogram of rock. |
| Luminous | A: The kilogram is a unit of mass, not weight. |
| OPT | A kilogram of metal is heavier than a kilogram of rock.
Why?
Because a kilogram of metal is heavier than a kilogram of rock. |

| What is heavier: A kilogram of metal or a kilogram of lead? | |
|---|---|
| GPT-3 | A kilogram of lead is heavier than a kilogram of metal. |
| Luminous | A: The kilogram is a unit of mass. It is defined such that 1 kg is the mass of the International Prototype Kilogram (IPK), which is a platinum-iridium international standard stored in France. |
| OPT | A kilogram of metal.
Why?
Because a kilogram of metal is heavier than a kilogram of lead. |

| "A kilogram of metal is heavier than a kilogram of feathers" is what most people say, but in reality | |
|---|---|
| GPT-3 | they weigh the same. |
| Luminous | , the weight of a kilogram of feathers is more than a kilogram of metal.
The reason is that the density of a substance is the mass of a substance divided by its volume. |
| OPT | , a kilogram of feathers is heavier than a kilogram of metal.
The reason is that a kilogram of metal is made up of atoms, and atoms are made up of protons, neutrons and electrons. |

| A wooden beam is placed over a stone. A small plastic bottle is placed on one side. What will happen if a person jumps onto the other side of the beam? | |
|---|---|
| GPT-3 | The beam will rotate and the person will jump into the air. |
| Luminous | A. The bottle will break.
B. The bottle will not break.
C. The bottle will break and the person will fall to the ground.
D. The bottle will not break and the person will fall to the ground |
| OPT | A wooden beam is placed over a stone. A small plastic bottle is placed on one side. What will happen if a person jumps onto the other side of the beam? |

| Mary can not move a heavy stone by herself. However, she brought a small object and a metal rod with her. Is Mary able to move the stone? | |
|---|---|
| GPT-3 | No, Mary is not able to move the stone. |
| Luminous | A: The answer is no.
The problem is that the metal rod is not a part of the stone.
The stone is a part of the object.
The object is a part of the system. |
| OPT | A. Yes
B. No
C. I don't know
D. I don't know |

| Mary can not move a heavy stone by herself. However, she brought a small object and a metal rod with her. How can Mary move the stone? | |
|---|---|
| GPT-3 | Mary can use the metal rod as a lever to move the stone. |
| Luminous | A: The answer is that Mary can move the stone by herself.
The problem is that the stone is too heavy for Mary to move by herself.
The solution is that Mary can move the stone by herself, but she *[TOKEN LIMIT]* |
| OPT | A. Mary can move the stone by using the rod.
B. Mary can move the stone by using the object.
C. Mary can move the stone by using *[TOKEN LIMIT]* |

| Mary can not lift a heavy stone by herself. However, she brought a small object and a metal rod with her. Is Mary able to lift the stone? | |
|---|---|
| GPT-3 | No, Mary is not able to lift the stone. |
| Luminous | A: The answer is no.
The problem is that the metal rod is not a part of the stone.
The stone is a part of the problem. |
| OPT | A. Yes
B. No
C. I don't know |

| | Mary can not lift a heavy stone by herself. However, she brought a small object and metal rod with her. How can Mary lift the stone? |
|---|---|
| GPT-3 | Mary can use the metal rod as a lever to lift the stone. |
| Luminous | A: The answer is that Mary can lift the stone if she can lift the rod. The rod is a lever. The force of the rod is the same as the force of the stone, but the distance the rod moves *[TOKEN LIMIT]* |
| OPT | Answer: Mary can lift the stone by using the rod. Explanation: Mary can lift the stone by using the rod. |

| | Mary can not lift a heavy stone by herself. However, she brought a calculator with her. Is Mary able to lift the stone? |
|---|---|
| GPT-3 | No, the calculator is not able to lift the stone. |
| Luminous | A: The answer is no. The calculator is not a machine, so it cannot lift the stone. |
| OPT | Mary can lift the stone. |

| | Mary can not lift a heavy stone by herself. However, she brought a calculator with her. How can Mary lift the stone? |
|---|---|
| GPT-3 | Mary can't lift the stone by herself, but she can use the calculator to help her. |
| Luminous | A: Mary can lift the stone by herself if she can calculate the force she needs to apply to the stone. |
| OPT | Answer: Mary can lift the stone by using the calculator. Explanation: Mary can lift the stone by using the calculator. |

## A.2 CAUSAL CHAINS QUESTIONS

| | If A causes B and B causes C. Does A cause C? |
|---|---|
| GPT-3 | It is possible that A causes C, but it is also possible that A and C are unrelated. |
| Luminous | A: The answer is no. The cause of a thing is the thing that makes it happen. |
| OPT | If A causes B and B causes C, then A causes C. If A causes B and B causes C, then A does not cause C. If A causes B and B does not cause C, then A does not cause C. |

| | If A causes B and B causes C. Does A cause B? |
|---|---|
| GPT-3 | There is not enough information to answer this question. |
| Luminous | A: The answer is no. The cause of a thing is the thing that makes it happen. |
| OPT | If A causes B and B causes C, then A causes C. If A causes B and B does not cause C, then A does not cause C. |

| | If A causes B and B causes C. Does B cause C? |
|---|---|
| GPT-3 | Yes, B causes C. |
| Luminous | A: The answer is no. The cause of a thing is the thing that makes it happen. |
| OPT | If A causes B and B causes C, then A causes C. If A causes B and B causes C, then A does not cause C. If A causes B and B does not cause C, then A does not cause C. |

| | If A causes B and B causes C. Does A cause A? |
|---|---|
| GPT-3 | A does not cause A. |
| Luminous | A: The answer is no. The cause of a thing is the thing that caused it. |
| OPT | If A causes B and B causes C, then A causes C. If A causes B and B causes C, then A does not cause C. If A causes B and B does not cause C, then A does not cause C. |

| If A causes B and B causes C. Does B cause A? | |
|---|---|
| GPT-3 | There is no definitive answer to this question as it depends on the specific situation. In some cases, B may cause A, while in others A may cause B. |
| Luminous | A: The answer is no.
The cause of A is A itself.
The cause of B is A and B.
The cause of C is A, B and C. |
| OPT | If A causes B and B causes C, then A causes C.
If A causes B and B causes C, then A does not cause C.
If A causes B and B does not cause C, then A does not cause C. |

| If A causes B and B causes C. Does C cause A? | |
|---|---|
| GPT-3 | There is no definitive answer to this question as it depends on the specific situation. In some cases, it may be true that C causes A, while in other cases it may not be true. |
| Luminous | A: The answer is no.
The cause-effect relationship is a logical relationship. It is not a physical relationship. |
| OPT | If A causes B and B causes C, then A causes C.
If A causes B and B causes C, then B causes C.
If A causes B and B causes C, then A does not cause C. |

| If A causes B, B causes C and C causes D. Does A cause D? | |
|---|---|
| GPT-3 | A causes D. |
| Luminous | A: The answer is no.
The cause of A is A itself.
The cause of B is A and B.
The cause of C is A, B and C. |
| OPT | If A causes B, B causes C and C causes D. Does A cause D?
If A causes B, B causes C and C causes D. Does A cause D? |

| If A causes B, B causes C, C causes D and D causes E. Does A cause E? | |
|---|---|
| GPT-3 | A causes E. |
| Luminous | A: The answer is no.
The cause-effect relationship is a logical relationship. It is not a physical relationship. |
| OPT | If A causes B, B causes C, C causes D and D causes E. Does A cause E?
If A causes B, B causes C, C causes D and D causes E. Does A cause E |

| If A causes B, B causes C, C causes D, D causes E, E causes F. Does A cause F? | |
|---|---|
| GPT-3 | A does not necessarily cause F. |
| Luminous | A: The answer is yes.
The cause of A is B, the cause of B is C, the cause of C is D, the cause of D is E, the cause of E is F. |
| OPT | If A causes B, B causes C, C causes D, D causes E, E causes F, then A causes F.
If A causes B, B causes C, C causes D, |

| If A causes B, B causes C, C causes D, D causes E, E causes F. Does B cause E? | |
|---|---|
| GPT-3 | There is no definitive answer to this question as the relationship between B and E could be indirect. |
| Luminous | A: The answer is no.
The cause of B is A, and the cause of A is B. |
| OPT | If A causes B, B causes C, C causes D, D causes E, E causes F. Does B cause E?
===Example=== |

| If A causes B, B causes C, C causes D, D causes E, E causes F. Does E cause B? | |
|---|---|
| GPT-3 | There is no definitive answer to this question as it depends on the specific circumstances of the situation. However, it is possible that E could cause B if, for example, E is a necessary component for the functioning of B. |
| Luminous | A: The answer is no. The cause of B is A, and the cause of A is B. The cause of C is B, and the cause of B is A. |
| OPT | If A causes B, B causes C, C causes D, D causes E, E causes F. Does E cause B? |

| If B causes C and A causes B. Does A cause C? | |
|---|---|
| GPT-3 | There is not enough information to determine if A causes C. |
| Luminous | A: The answer is no. The cause of A is B, and the cause of B is A. The cause of C is A, and the cause of A is B. |
| OPT | If A causes B, then A causes C. If A causes B, then C causes B. If A causes B, then C causes A. |

| If B causes C and A causes B. Does C cause A? | |
|---|---|
| GPT-3 | There is no definitive answer to this question as it depends on the specific situation. In some cases, it may be true that C causes A, while in other cases it may not be true. |
| Luminous | A: The answer is no. The cause of A is B, and the cause of B is A. The cause of C is A, and the cause of A is B. |
| OPT | If A causes B, then C causes A. If A causes C, then B causes A. |

| If G causes Q and Q causes S. Does G cause S? | |
|---|---|
| GPT-3 | Yes, G causes S. |
| Luminous | I'm trying to understand the concept of causation. I'm not sure if I'm understanding it correctly. If G causes Q and Q causes S. Does G cause S? A: The answer is no. |
| OPT | If G causes S, then G causes Q. If G causes Q, then G causes S. If G does not cause S, then G does not cause Q. If G does not cause Q, then G does not cause S |

| If Q causes S and G causes Q . Does G cause S? | |
|---|---|
| GPT-3 | Yes, G causes S. |
| Luminous | A: The answer is no. The reason is that the cause of S is Q, and the cause of Q is G. So, the cause of S is G. |
| OPT | If Q causes S and G causes Q, then G causes S. If Q causes S and G causes Q, then G does not cause S. If Q causes S and G does not cause Q, then G does not cause S. |