# OpenReview forum: "Can Foundation Models Talk Causality?"
_auai.org/UAI/2022/Workshop/CRL — CRL@UAI 2022 Poster_

### Official Review · Reviewer_wgE5 · 2022-06-23
**The paper discusses an interesting research topic of using language model to uncover causality. The contribution seems to be limited.**

**Rating:** 5
**Confidence:** 3

**Review:**

The paper examines to what extent the fundamental language models, namely, Luminous, GPT3 and OPT, can talk about causality. By answering the question concerning "is there a correlation" and "is there a causation", the models are tested on various scenarios and some interesting results are reported.

I like the idea of exploring causality from the language models. This is a novel and interesting research direction. I have several concerns considering the current version of the paper.

1. It seems that the paper is experimental only. The authors may consider to answer more question or provide more insights about the problems like "is there any theory that can explain the results?" "where does the difference between multiple language model come from?" "How reliable the current shown results are?". If intended to do conduct research on experimental level, a more systematical experiments are better and would be more appreciated.

2. Some important experimental details are lacking. It is known that performance of AI models are sometimes sensitive to the model architecture, parameters such as learning rates etc. The authors should put down more details so that the readers can better interpret the results. It would also be better to compare the differences of the language models on architecture level, so that more insights of where the difference come from and how much causal information is captured, can be provided. It would be also useful to show if variance of the performance exist, by conducting experiments using different level of randomness.

---

### Official Review · Reviewer_9mfD · 2022-06-28
**Comments for the authors**

**Rating:** 6
**Confidence:** 4

**Review:**

Summary: This paper considers whether foundation models (FMs), large models pre-trained on immense amounts of data, encode knowledge that allows them to draw causal inferences. This topic has come under debate and scrutiny of late. In this context, the authors propose a hypothesis about what FMs encode and study it empirically. This paper hypothesizes that even though FMs are not trained explicitly parametrized with causal models, some components of FMs maybe encoding causal structure assumptions, such as a common cause structure among variables. Then, this causal structure may be being used in inferences such as answering natural language questions like "does smoking cause cancer?" The authors design studies using a variety of causal discovery and commonsense reasoning benchmarks to evaluate this hypothesis about FMs.

Strengths:
+ The paper is ambitious in the topic it addresses and the hypothesis is proposes. Understanding what FMs encode and poking them in ways to elicit what causal information they encode is a very new topic but an important one, and arguably, a very challenging one. The difficulties are inherent in a field that has to draw conclusions about black boxes by only varying inputs to the black box and evaluating the outputs. As such, proposing what seems to me like a novel hypothesis about FMs encoding structural assumptions is ambitious in its  reach and scope.
+ The empirical studies are clearly described, with research questions articulated.
+ The empirical studies are comprehensive in that they cover 3 LMs, multiple tasks, and evaluate sensitivity of the results to various parameters of the studies.

Some concerns and comments for the authors:
+ Consider the chocolate, Nobel laureates and GDP example in Figure 1. The meta-level causal assumptions hypothesis is motivated by the idea that from lots of training data that contain natural language sentences about these concepts, the model might somehow encode the shown common cause structure. But I feel that there's a gap in explaining how FMs could possibly do this using mechanisms like attention that are purely associational and can't capture the asymmetry of causality. For example, the FM may just learn to make the embeddings for GDP, Nobel laureates and chocolate production close in embedding space. I think the authors could provide more technical justification (doesn't have to be super formal) or reasoning behind why their hypothesis is a plausible one, given what we know about FMs and their parameters.

+ One big concern about the empirical studies is that the evaluation is only based on outputs of a LLM. The studies don't involve peering into the black box here to see which components of the model are responsible for the answers to the queries about causality. In the absence of knowing what model components were responsible for a correct answer, how can we distinguish whether the correct inference was due to meta-level causal assumption or just memorization of the pattern from enough training data.

+ In general, I'd encourage the authors to think more about designing empirical studies that could rule out alternative hypotheses for why an FM might be able to correctly answer a "does X cause Y" question or a physics/commonsense question. It would be much stronger evidence for the causal structural assumptions hypothesis if other explanations such as memorization could be more convincingly ruled out. As it stands, I think the empirical studies reveal lots of interesting artifacts, especially about FM-G's generally good performance. However, I'm not sure if we can really say anything about the hypothesis that FM-G's performance is good *because* it encodes some causal assumptions.

---

### Meta-Review · Program_Chairs · 2022-07-05

**Recommendation:** Accept (Poster)
**Confidence:** 3

**Metareview:**

Both reviewers agree that the problem the paper is tackling is interesting, but that the paper might still need some work. Since this is a workshop and we are interested in discussing/developing new ideas, we recommend acceptance for this paper. On the other hand, we would suggest that the authors take into account the concerns from the reviewers, especially in terms of designing empirical studies and ablations studies to rule out alternative explanations.

---

### Decision · Program_Chairs · 2022-07-06

Accept (Poster)